# Multi-Scenario Analysis of Habitat Quality in the Yellow River Delta by Coupling FLUS with InVEST Model

**DOI:** 10.3390/ijerph18052389

**Published:** 2021-03-01

**Authors:** Qinglong Ding, Yang Chen, Lingtong Bu, Yanmei Ye

**Affiliations:** 1Land Academy for National Development, Zhejiang University, Hangzhou 310029, China; qinglongding1212@gmail.com (Q.D.); herochen945@163.com (Y.C.); 2School of Business, Nanjing University, Nanjing 210093, China; kookieplt@163.com

**Keywords:** habitat quality, multi-scenario analysis, FLUS, InVEST model, the Yellow River Delta

## Abstract

The past decades were witnessing unprecedented habitat degradation across the globe. It thus is of great significance to investigate the impacts of land use change on habitat quality in the context of rapid urbanization, particularly in developing countries. However, rare studies were conducted to predict the spatiotemporal distribution of habitat quality under multiple future land use scenarios. In this paper, we established a framework by coupling the future land use simulation (FLUS) model with the Intergrated Valuation of Environmental Services and Tradeoffs (InVEST) model. We then analyzed the habitat quality change in Dongying City in 2030 under four scenarios: business as usual (BAU), fast cultivated land expansion scenario (FCLE), ecological security scenario (ES) and sustainable development scenario (SD). We found that the land use change in Dongying City, driven by urbanization and agricultural reclamation, was mainly characterized by the transfer of cultivated land, construction land and unused land; the area of unused land was significantly reduced. While the habitat quality in Dongying City showed a degradative trend from 2009 to 2017, it will be improved from 2017 to 2030 under four scenarios. The high-quality habitat will be mainly distributed in the Yellow River Estuary and coastal areas, and the areas with low-quality habitat will be concentrated in the central and southern regions. Multi-scenario analysis shows that the SD will have the highest habitat quality, while the BAU scenario will have the lowest. It is interesting that the ES scenario fails to have the highest capacity to protect habitat quality, which may be related to the excessive saline alkali land. Appropriate reclamation of the unused land is conducive to cultivated land protection and food security, but also improving the habitat quality and giving play to the versatility and multidimensional value of the agricultural landscape. This shows that the SD of comprehensive coordination of urban development, agricultural development and ecological protection is an effective way to maintain the habitat quality and biodiversity.

## 1. Introduction 

The global is experiencing unprecedented habitat quality degradation since most countries move up the ladder of urbanization, which results in biodiversity degradation [1,2]. Habitat quality refers to the ability of the environment to provide suitable conditions for biological survival and development [3,4]. This reflects the integrity and diversity of regional ecosystem function to a certain extent. However, dramatic land use change may trigger the decline of habitat quality [2,5]. Therefore, authorities, planners, and scholars reached a consensus on preserving habitat quality in the context of rapid urbanization. For example, the European Commission and the United States have formulated biodiversity conservation strategies and bills to prevent further deterioration of the ecological environment. China planned 23% of its territory as a preferential conserved area for biodiversity conservation [6]. The 15th Conference of the Parties to the Convention on Biological Diversity pointed out that stable habitat quality is an important foundation for maintaining ecosystem biodiversity and will determine a new global biodiversity target for 2030 [7]. Under such circumstances, it is of great significance to analyze the spatiotemporal change in habitat quality to support environmental protection.

In recent decades, land use change induced by rapid urbanization across the globe imposed a profound impact on habitat quality. For example, Lawler et al. [8] found that projected land-use changes will result in >10% decreases in habitat for 25% of modeled species in the United States in 2051. Particularly, urban expansion affects habitat quality, leading to habitat fragmentation, water quality degradation, and freshwater scarcity [9,10]. Previous literature showed that future urban expansion would lead to more fragmented wetland landscapes and the degradation of habitat quality, which can be alleviated by limiting urban areas [10]. Unfortunately, the global urban land has increased by 58,000 km^2^ from 1970 to 2000 [11]; the amount has been projected to increase by 1.2 million km^2^ from 2000 to 2030 under current trends [12]. This indicated that the threats of the continuous global urban expansion on habitat quality will be continued for several decades. Therefore, projecting the consequences of land use change on habitat quality is an urgent task for coordinating sustainable land development and habitat conservation.

An increasing body of literature has focused on assessing spatiotemporal pattern of habitat quality [2,13,14], the impact of land consolidation on habitat quality [2,15,16,17,18], the relationship between habitat quality and landscape pattern [13,19], and the trade-off between habitat quality and other ecosystem services [20,21,22]. In terms of study areas, there are a plethora of studies focusing on urban areas, watersheds, nature reserve, etc. [15,23,24]. Although existing literature paid more attention to assessing the impact of past/current land use change on habitat quality, rare attempts were conducted to project the spatiotemporal distribution of habitat quality under the future land use scenarios [25]. Meanwhile, little attention was put to eco-sensitive areas. 

Yellow River Delta (YRD) is a typical area whose terrestrial ecosystem, aquatic ecosystem, and estuary wetland ecosystem are intertwined. The land is highly salinized, the freshwater resources are scarce, and the resource carrying capacity is weak. The area is an important functional area for grain production and an economic development zone, with a population of more than 10 million. YRD is experiencing a contradiction between habitat conservation and land development. On the one hand, rapid urbanization and the consequent land use change are prevalent in the Yellow River Delta. At the beginning of 21st century, the local government planned the Yellow River Delta Efficient Ecological Economic Zone as an engine for economic growth. For example, the built-up land was increased from 540 km^2^ in 1995 to 2417.42 km^2^ in 2015 in Dongying City [26]. On the other hand, the Yellow River Delta has attracted attention to ecological conservation in the context of emphasizing eco-civilization. The Yellow River Delta provides a hotbed for endangered birds, reptiles, zoobenthos, etc. [27]. Given the important role for rare species survival, the Yellow River Delta was approved as a National Nature Reserve. In this regard, the contradiction between habitat conservation and land development raised the concern about whether future land development affect the habitat quality in the Yellow River Delta.

To address the gap, we projected the spatiotemporal change of habitat quality under four land use scenarios: business as usual (BAU), fast cultivated land expansion scenario (FCLE), ecological security scenario (ES) and sustainable development scenario (SD) in the Yellow River Delta using Future Land Use Simulation model (FLUS) and the Intergrated Valuation of Environmental Services and Tradeoffs (InVEST) model. The main goals of the present work include: (1) simulating future land use and habitat quality under four scenarios in 2030, and (2) comparing spatiotemporal change of the habitat quality. This study can be applied to support the spatial planning for an ecological sensitive region.

## 2. Material and Methods 

### 2.1. Study Area 

Dongying Cityis one of the most important cities in the Yellow River Delta High-efficient Eco-Economic Zone (YRDHEZ) (Figure 1). Dongying is located at mid-latitude and belongs to the warm temperate continental monsoon climate. The annual average temperature, the accumulated temperature ≥10 ℃, and the annual average precipitation are 12.8 ℃, 4300 ℃, and 555.9 mm, respectively. A large number of sediments carried by the Yellow River are deposited along the Yellow River Estuary in Dongying City, forming large-scale terrestrial ecosystems, aquatic ecosystems and estuary wetland ecosystems.

The area is abundant with species: 367 bird species in the territory, accounting for 21% of the total bird species in the country; 641 aquatic animals, including 108 freshwater fish and 85 marine fish [28]. However, due to population growth and rapid urbanization, the local ecosystem is facing considerable pressure. In addition, the region has a large amount of arable land reserve resources, and agricultural reclamation poses a threat to habitat quality. The built-up land was increased by 347.34% during 1995 and 2015 [26]. Natural factors and land use change have resulted in loss of waterfowl habitat and biological diversity in this area [29,30]. 

Multi-source data, including land use, planning, terrain, accessibility, climate, and socioeconomic data, was used (Table 1). Land use data in 2009 and 2017 was obtained from Dongying Natural Resources Bureau. Planning data, including ecological redline and permanent primary farmland, was extracted from the land use planning of Dongying City (2020–2035). Digital elevation model (DEM) was obtained from the Resource and Environment Science Data Center of Chinese Academy of Sciences. Slope and aspect data were extracted from DEM using slope analysis in ArcGIS. Built up area, town, road network, and river were obtained from the Geographical Information Monitoring Cloud Platform. Accessibility data was extracted by using Euclidean distance. Meteorological data was obtained from the China Meteorological Administration (Figure 2). Population and GDP with a resolution of 1 km were collected from the Resource and Environment Science Data Center of Chinese Academy of Sciences.

### 2.2. Methodology 

The methodology in this paper includes three steps (Figure 3). The first step is setting future land use scenarios. According to regional planning policy including ecological red line and permanent basic farmland, we defined four scenarios, namely BAU, FCLE, ES and SD. Markov chain was used to predict the amount of land use demand in 2030. The second is simulating future land use changes using FLUS model under four scenarios. The third step is assessing and comparing spatiotemporal distribution of habitat quality in 2009, 2017, and 2030 using InVEST model.

#### 2.2.1. Defining Scenarios and Calculating Future Land Use Demand

It is necessary to consider the constraints to control the intensity of urban expansion and agricultural development in future land use simulation. So, we adjusted the conversion probability, intensity and direction between different land types by adding the planning restricted area and setting parameters. In this way, the multi-scenario simulation can reflect the development under planning constraints. For example, under the fast-cultivated land expansion scenario, the probability that other land types transferring to the cultivated land should be appropriately increased. Under the ecological protection scenario, the occupation of ecological land by other land types should be reduced, and the restricted areas should be delimited. We set four scenarios by combining land use needs, restricted areas, neighborhood factors, and conversion costs (Table 2, Table 3, Table 4 and Table 5). The S1 is the business as usual (BAU) scenario. According to the land use transfer matrix from 2009 to 2016, the Markov chain is applied to forecast land use demand in 2030. The parameters of the FLUS model under BAU remain unchanged. The S2 is the fast-cultivated land expansion (FCLE) scenario in which, cultivated land was reduced from 235,032 hm^2^ to 233,273 hm^2^; water body, wetland, construction land, and unused land were reduced from184,891 hm^2^, 116,307 hm^2^, 129,437 hm^2^, 125,779 hm^2^ to 181,474 hm^2^, 112,133 hm^2^, 169,957 hm^2^, 95,849 hm^2^ during 2017 and 2030; the transfer probability of construction land was further adjusted. The S3 is the ecological security (ES) scenario, in which construction land, water body, and wetland were increased to 159,390 hm^2^, 187,215 hm^2^, and 116,757 hm^2^; cultivated land, and unused land were reduced to 220,313 hm^2^ and 107,707 hm^2^. Ecological redline was defined as the restricted area. In terms of FLUS, the neighborhood factors were adjusted; the conversion cost of ecological land types were increased. The S4 is the sustainable development (SD) scenario, in which construction land, and water body were increased to 159,390 hm^2^, 186,420 hm^2^; cultivated land, wetland and unused land were reduced to 225,065 hm^2^, 116,179 hm^2^ and 104,743 hm^2^. Taking into consideration urban expansion, arable land reclamation and ecological protection, the area of land demand and expansion capacity are adjusted, and the ecological redlines and permanent primary farmland are added to the simulation model as restricted areas. The land transfer cost is consistent with S3.

The Markov model assumes that the status of land use at *t + 1* is according to the former status of land use at *t*, which predicts the future land use transition probability according to the current land use transition probability. The formula is as follows:(1)$$S_{t+1}=P_{ab}\times S_{t}$$
where $S_{t}$, $S_{t+1}$ is the matrix of land use status in the study area at time *t* and *t +* 1. $P_{ab}$ is the transition probability matrix for the conversion of land type *a* to land type *b*.

#### 2.2.2. Simulating Future Land Use Using FLUS Model 

The FLUS model is composed of the artificial neural network (ANN) and the adaptive inertia competition mechanism. The ANN is effective to discover the relationship between the natural, social, and economic elements and land use change. The adaptive inertia competition mechanism is useful to crack the uncertainty and complexity of mutual transformation, which can remedy the complexity of local conversion and parameter determination in traditional cellular automata. As the natural environment, population density and economic development jointly drive the land use change, this paper selected 15 driving factors such as the terrain, accessibility, average temperature as the influencing factors of the FLUS model (Table 1). The FLUS model has better prediction ability and higher accuracy than the CLUE-S model [31,32,33].

##### Calculation of Probability-of-Occurrence Estimation Using Artificial Neural Networks

Artificial neural network (ANN) is an effective tool to iterate, adjust and fit the relationship between the input data and the training target through learning-recall. ANN was proven to be an easier way to deal with the complex and nonlinear relationship between land use and multiple driving factors [31,34,35].
(2)$$sp\left(p,k,t\right)=\underset{j}{\sum }\omega_{j,k}\times sigmoid\left\{net_{j}\left(p,t\right)\right\}=\frac{\sum_{j}\omega_{j,k}}{1+e^{-net_{j}\left(p,t\right)}}$$

In this paper, the driving factors include terrain, accessibility, socioeconomic and climate factors (Table 1). Where $\omega_{j,k}$ is an adaptive weight between the hidden layer and the output layer, and it is adjusted during the training process. $net_{j}\left(p,t\right)$ denotes the signal received by the neuron *j* in the input layer at pixel *p* at training time t. sigmoid () is the excitation function from the hidden layer to the output layer. For the suitability probability sigmoid () output by ANN, at the iteration time *t* pixel *p*, the sum of suitability probabilities of various types of land is constant to 1, that is:(3)$$\underset{k}{\sum }sp\left(p,k,t\right)=1$$

##### Self-Adaptive Inertia and Competition Mechanism 

The inertia coefficient is the core of the self-adaptive inertia competition mechanism. It is affected by the land quantity and the land use demand. It means that when the development trend of a certain land use type does not meet the demand, the inertia coefficient will adjust and correct itself in the next iteration, thus making all kinds of land use quantity evolves to the set land demand [36]. The inertia coefficient is as follows:(4)$$Inertia_{p}^{t}=\{\begin{array}{c} Inertia_{p}^{t-1}\;\;\;\;\;\;\;\;\;\;\;\;\;\;\;\;|D_{p}^{t-2}|\leq |D_{p}^{t-1}|\\ Inertia_{p}^{t-1}\times \frac{D_{p}^{t-2}}{D_{p}^{t-1}}\;\;\;\;\;\;0>\;D_{p}^{t-2}>\;D_{p}^{t-1}\\ Inertia_{p}^{t-1}\times \frac{D_{tp}^{t-1}}{D_{p}^{t-2}}\;\;\;\;\;\;\;\;\;D_{p}^{t-1}>\;D_{p}^{t-2}>0 \end{array}$$
where $Inertia_{p}^{t}$ represent the inertia coefficient for land use type *p* at iteration time *t*. $D_{p}^{t-1}$, $D_{p}^{t-2}$. refers to the difference between land use grid allocation and macro demand of land use type *p* at time *t* − 1, *t* − 2. 

Base on the probability-of-occurrence, neighborhood effect, inertia coefficient and conversion cost, the FLUS model established a more comprehensive probability using the following equation:(5)$$TProb_{k,p}^{t}=SP\left(k,p,t\right)\times \Omega_{k,p}^{t}\times Inertia_{p}^{t}\times \left(1-sc_{c\to p}\right)$$
where SP(k,p,t) represents the probability-of-occurrence of land use type *p* on grid cell *k* at time *t*;
$sc_{c\to p}$ represents the conversion cost from the land use type *c* to *p*; $\Omega_{k,p}^{t}$. represents the neighborhood effect of land use *p* at a specific grid cell *k*, the equation is defined as: (6)$$\Omega_{k,p}^{t}=\frac{\sum_{N\times N}con\left(c_{k}^{t-1}=p\right)}{N\times N-1}\times \omega_{p}\;$$
where $\sum_{N\times N}con\left(c_{k}^{t-1}=\;p\right)$ denotes the total number of grid cells occupied by the land use type *p* at time *t-1* within the *N*×*N* window, *N* = 3 in this paper. $\omega_{p}$ denotes the weight of neighborhood effect of various type of land use *p*.

#### 2.2.3. Assessing Habitat Quality Using InVEST Model 

The disappearance and degradation of natural habitat is the main cause of biodiversity loss [1,37,38]. Habitat quality is the ability and potential of environment to provide suitable conditions for survival and reproduction of organisms [2,14,39]. In the InVEST model, we assume that biodiversity is proportional to habitat quality. In order to provide land managers with the spatial distribution of biodiversity in each patch, the habitat quality of the study area was evaluated dynamically. Habitat quality was calculated by given equation as:(7)$$Q_{xj}=H_{j}\left[1-\left(\frac{D_{xj}^{z}}{D_{xj}^{z}+k^{z}}\right)\right]$$
where $Q_{xj}$. denotes the habitat quality index of grid unit x of landscape type j; $H_{j}$. denotes the habitat suitability score of the landscape j, ranging from 0 to 1; *z* is the scale constant, which is generally taken as 2.5; *k* is the semi saturated constant, and we took it as 0.5; $D_{xj}^{Z}$. denotes the habitat degradation index, which indicates the degree of habitat degradation under stress, the formula is as follows:(8)$$D_{xj}=\overset{R}{\underset{r=1}{\sum }}\overset{Y_{r}}{\underset{y=1}{\sum }}\left(\omega_{r}/\overset{R}{\underset{r=1}{\sum }}\omega_{r}\right)r_{y}i_{rxy}\beta_{x}S_{jr}\;$$
where *R* is the number of stress factors; $Y_{r}$. is the total number of grid units of stress factors; $\omega_{r}$ is the weight; $r_{y}$ is the number of stress factors on the grid unit; $\beta_{x}$ is the accessibility level of grid x (the legal protection level, such as strict protection area is taken as 1, and the harvest type protection is taken as 0, otherwise the value is between 0 and 1.); $S_{jr}$. denotes the sensitivity of landscape j to stress factors, 0–1; $i_{rxy}$ denotes the influence distance of stress factors, including linear and exponential decline, and the calculation formula is as follows:(9)$$i_{rxy}=1-\left(\frac{d_{xy}}{d_{r\;\;max}}\right)\;\;if\;\;linear$$
(10)$$i_{rxy}=exp\left(-\left(\frac{2.99}{d_{r\;\;max}}\right)d_{xy}\right)\;\;if\;exponential\;$$
where $d_{xy}$ is the linear distance between grid *x* and *y*; $d_{r\;max}$ is the maximum operating distance of the threat *r*. 

This module needs land use, stress factors, stress sources, habitat types, and the sensitivity and semi saturation parameters of habitat types to stress. In this paper, the relevant parameters are determined by combining existing studies and the status of the Yellow River Delta [14,16,40]. The Yellow River Delta National Nature Reserve is taken as the protection zone, and the accessibility is assigned as 1. See Table 6 and Table 7 for the specific influence distance, weight and sensitivity of land use to habitat threats.

## 3. Results Analysis

### 3.1. Multi-Scenario Simulation for Future Land Use

Research showed that the closer the overall and kappa coefficient are to 1, the better the simulation accuracy is, and vice versa [41]. When the kappa coefficient is greater than 0.75, the simulation accuracy is therefore realiable and has statistical significance [31,36]. The verification results showed that the overall and the kappa coefficient are 0.85 and 0.81, respectively. The experimental accuracy reached a high level, indicating that the FLUS model has good applicability and can be applied to the future multi scenario simulation in 2030.

Figure 4 shows the multi-scenario simulation for future land use under four scenarios in 2030. Under the BAU (Figure 4S1), urban expansion is evident, especially in the central city and along the Yellow River. Under the FCLE (Figure 4S2), while urban expansion in the central city was restrained, that in the northeast of Dongying is more evident than BAU. Cultivated land is well-preserved, as revealed by the proportion of cultivated land decreased by 0.21% from 2017 to 2030 (Figure 5). Under the ES (Figure 4S3), the expansion of cultivated land has suppressed the encroachment of wetlands and waterbody in the south and east of Dongying. Under the SD (Figure 4S4), the expansion of construction land around the central city has been suppressed, showing infill and compact expansion. The cultivated land in the south of the city has been better protected, and so as to the unused land in the east and on the both sides of the Yellow River. 

### 3.2. Habitat Quality Assessment under Different Scenarios

The spatial distribution of habitat quality in Dongying City in 2009 and 2017 is shown in Figure 6. The high-quality habitat is mainly distributed in the Yellow River Estuary and water area, while the low- quality habitat is located in the central city and southern areas (Figure 6). Since the implementation of the land use planning, the low-quality habitat areas have gradually expanded in the middle of Dongying District and around the mining land in the northern oil fields. Statistical analysis shows that the average of habitat quality in 2009 and 2017 are 0.4172 and 0.4100, respectively. It indicated that the overall habitat quality still shows a downward trend and the standard deviation of the habitat quality index rose from 0.2745 to 0.2810, showing that the spatial fluctuation range of the habitat quality between grid cells was increasing, and the difference in habitat quality was expanding.

Under the four different scenarios, habitat quality showed a rapid growth, but different trends. Among them, the average value of habitat quality under the SD (0.4482) was the largest, and the BAU (0.4377) was the smallest (Figure 7). It can be observed that the overall pattern of habitat quality in Dongying City has not changed greatly in the four scenarios of 2030 (Figure 8). The areas with higher-grade habitat quality are mainly distributed in the northern and eastern coastal areas of Dongying City, as well as the Yellow River water surface. The lower-grade habitat quality is mainly distributed in urban center and coastal saline alkali areas. The area of middle-grade habitat quality is small and distributed in the Yellow River Estuary.

### 3.3. Comparison of Habitat Quality under Four Scenarios

In this paper, the habitat quality index was divided into five grades, and the area proportion of each grade in 2017 and 2030 was calculated (Table 8). We compared the habitat quality grades in 2017 with that in four scenarios, and found that the pattern of the five habitat quality grades changed significantly. In 2017, the largest proportion of habitat quality area was low-grade, followed by lower-grade and high-grade, accounting for 30.07%, 26.99%, 26.55%. The higher-grade quality habitat accounted for 7.03%, and this ratio exceeded 23.99% in 2030. Under the BAU, the area proportion of lower-grade was the largest, accounting for 31.34%, followed by low-grade, and higher-grade, accounting for 28.41% and 23.99% respectively, while the area of middle-grade, was the smallest, accounting for 1.51%. Under the FCLE, the proportion of lower-grade area decreased to 29.14%, and the proportion of higher-grade, area increased to 24.70%. Under the ES, the areas of medium, high and higher grade were 1.73%, 14.8% and 24.51% respectively. Under the SD, the habitat quality of middle and higher grade increased significantly, which were 1.68% and 24.53% respectively, while the proportion of lower- grade habitats decreased to 29.67%. In general, the habitat quality grade in 2030 changed a lot compared with that in 2017. The proportion of higher grade improved greatly, and the proportion of middle grade and high grade reduced greatly. According to the four scenarios, the largest proportion of the sum of the high and higher grade is the SD, followed by ES, FCLE and BAU, accounting for 39.33%, 39.31%, 38.99%, 38.73% (Table 8).

From the perspective of spatial distribution change, areas with increased habitat quality are more concentrated and contiguous. The added value is mainly between 0.1 and 0.4, which is mainly distributed in the northern and eastern regions. The areas where habitat quality decreased were scattered, and the decrease values were mainly between −0.4 and −0.1 (Figure 9). The habitat quality index increased by more than 0.4 in the southeast part of the BAU (Figure 9S1). Under the FCLE, the decrease values of habitat quality were lower than -0.4 in the central and southeast parts of the FCLE (Figure 9S2). In the four scenarios, the areas where the habitat quality index decreased less than -0.4 were basically situated in the Yellow River Estuary. Generally speaking, the areas where the habitat quality of Dongying City is facing the risk of degradation are mainly caused by the change of water area and wetland. The key and difficult points of biodiversity protection are still located in the Yellow River Delta National Nature Reserve. Compared with the BAU and FCLE, the areas with decreased habitat quality were more concentrated in the ES and SD scenarios, mainly in the eastern, northern and urban surrounding areas (Figure 9S3, S4).

## 4. Discussion 

### 4.1. Future Land Use and Habitat Quality

Four scenarios will experience a dramatic change in both the amount and spatial patterns of each land use type. The BAU showed a higher urban expansion around the central urban area. The FLCE disclosed the lower urban expansion, and highest agricultural expansion. The ES exhibited the lowest urban expansion and agricultural expansion. The SD showed moderate urban expansion, and cultivated land expansion. Habitat quality showed a downward trend during 2009–2017, which is consistent with the findings that the ecosystem service emergy declined [42]. And it is also consistent with the decline of the shorebird habitat quality in the YRD [29]. The main reason is that urban expansion invaded the cultivated land, forest, wetland and grassland. The occupation of cultivated land by the construction land, mostly occurs around the central urban areas.

From the regional perspective, the area where the habitat quality decreased by more than 0.4 under the BAU were mainly located in the estuary of the Yellow River, the north and the surrounding areas of the central urban area. Under the BAU, the areas with habitat quality rising more than 0.4 were mainly located in the eastern and northern of the central urban area. In contrast, the areas where the habitat quality declined by more than 0.4 were mainly in the estuary, along the Yellow River and southeast of the city under FCLE. That is because the wetland and forest were transferred to the cultivated land in those areas. The decreasing and increasing trend of habitat quality under the ES was similar to that under the BAU, but the decreasing intensity became weaker, especially in the estuary of the Yellow River. Differently, the decreasing areas is less and the increasing area is more under the SD. Therefore, the habitat quality in the estuary of the Yellow River is under great pressure of degradation. Under the FCLE the overall habitat quality in Dongying city is better, yet the coastal area of the Yellow River is facing seriously degradation of habitat quality.

### 4.2. Impacts of Future Land Use on Habitat Quality

Urban expansion can easily cause the rapid decrease of land with important natural habitat functions, such as waters, wetland, and forests, resulting in a decline in habitat quality [43,44]. Studies have shown that the decline of biodiversity and the decline of habitat quality are mostly related to urban expansion. From 2009 to 2017, the study area was in a stage of rapid urbanization, and urban expansion had a serious occupation of natural habitats. Urban construction land and transportation land are considered to be the main threat sources. On the one hand, urban expansion has caused the fragmentation of complete natural habitats and blocked the transmission of material and information flows. On the other hand, environmental pollution caused by urban construction has also caused the disappearance and extinction of biological populations [17].

Agricultural expansion leads to a decline in habitat quality [45], but our research results seem that agricultural expansion has caused an increase in habitat quality. Why is that? Due to China’s unique policy of “the dynamic balance of cultivated land occupation and compensation” [46], that is, the amount of cultivated land occupied by construction must be increased as much as the same quality and quantity. The way to make up for the loss of the cultivated land is land reclamation. Different from urban construction land, cultivated land also has a certain ecosystem service function and can provide habitat for living things [47]. Compared with unused land such as saline-alkali land in the study area, cultivated land has better functions of soil conservation, water conservation and biodiversity protection. During the period of rapid urbanization, urban expansion and agricultural development are dominated by the occupation of cultivated land, wetlands, and water bodies. In the future, as more such land will be included in the ecological red line and permanent basic farmland protection, agricultural expansion will turn to land with weaker ecological functions, such as saline-alkali land. This kind of development may increase the amount of cultivated land in the region, as well as improve the soil environment and regulate the salt content, thereby providing a better habitat for animals and plants. However, it should be noted here that due to the lack of freshwater resources and the extremely uneven annual rainfall in the study area, large-scale agricultural development easily takes up too much freshwater resources, which will cause competition and conflicts in freshwater resources [48]. For example, the conflict between agricultural irrigation and ecological water supply in nature reserves, and the competition between agricultural irrigation and domestic water and industrial water use. If the problem of freshwater resources cannot be solved, large-scale agricultural development may improve the quality of habitat in the short term, but it is likely to cause greater ecological problems in the long run.

### 4.3. Policy Implications 

This study can provide policy implications for future land use according to the impact of different land use patterns on habitat quality. First, cultivated land and water body are widely distributed around the city. Therefore, the loss of cultivated land and water body plays a key role in the degradation of habitat quality. Land managers and planners should balance the relationship between urban expansion and ecological protection by focusing on improving land use efficiency and avoiding areas with high habitat quality. Secondly, appropriate development of saline alkali land and unused land as cultivated land will help to improve the habitat quality in the northern area, but the occupation of wetlands and water body should be avoided. Thirdly, there are a lot of woodland, grassland, water body and wetland in the estuary of the Yellow River, which have high habitat quality. On the one hand, agricultural development may occupy such area. On the other hand, there may be tradeoffs and competition between upstream agricultural water and downstream ecological water, and excessive agricultural activities in the upstream will bring environment pollution. The planners should delimit the core area of ecological redline and prohibit construction and agricultural activities. Fourth, agricultural development should pay more attention to strengthen the management of saline alkali land, improve the quality of cultivated land. Taking an intensive agricultural development mode is a more likely way of sustainable land use, rather than constantly encroaching on ecological land to expand the cultivated land area.

### 4.4. Limitations and Improvement 

Our research still has some limitations. The potential influence of land use on habitat quality depends largely on the predicted urban land use. Although the FLUS model combines human activities and natural factors, and it has higher simulation accuracy, but the model relies on the prediction of land use demand. However, the Markov chain is a model with a simple description of historical data and trend perdition, without considering the current situation of economic development and the promotion of local economic and social development on land use [49,50]. In addition, the parameter setting is subjective and fails to consider the differences of time and region in the habitat quality assessment based on InVEST model. Actually, the threats to habitat quality are complex. However, the habitat quality based on the model can only reflect its relative value. If there is more accurate species distribution, we can make a more detailed assessment [40]. 

## 5. Conclusions

Simulating the impacts of land use on habitat quality under multi-scenarios was widely conducted. Nevertheless, rare attempts were conducted to project the spatiotemporal distribution and change of habitat quality of future land use. The combination of the FLUS with InVEST model was found to be a useful tool for assessing the spatial distribution and change of habitat quality of future land use. In this paper, we established an analytical framework to explore the impact of future land use on habitat quality. Land use not only affects habitat quality, but also threatens other ecosystem services, such as carbon storage and water conservation. The conceptual framework established above can also help to analyze the impact of land use on other ecosystem services.

The main conclusions are as follows: (1) From the view of spatial evolution, the urban expansion under the BAU is evident, especially around the urban center and the western region of the Yellow River. The FCLE shows that the construction land expansion is restrained and the degradation rate of cultivated land slows down. Under the ES, the construction land expansion further slowed down, while the wetland, water and forest land showed a slight increase, but the cultivated land area decreased by 1.57% compared with the FCLE. The SD shows that the construction land expansion presents compact and filling expansion, and the change range of cultivated land, water area and wetland is small; (2) The assessment results show that the average habitat quality of Dongying City in 2009 and 2017 was 0.4172 and 0.4100, respectively. The standard deviation showed that the spatial distribution of habitat quality was uneven. The areas with high value were mainly located in the Yellow River Estuary and some coastal areas, and the low value areas were mainly distributed in the central and southern of Dongying City. The habitat quality distribution is with strong spatial heterogeneity. Under the four land use scenarios, the habitat quality was 0.4377, 0.4478, 0.4466 and 0.4482, respectively. The habitat quality index increased compared with that in 2017. From the perspective of spatial change, the areas with increased habitat quality were more concentrated, mainly situated in the eastern and northern regions, and the areas with lower habitat quality were scattered and overlapped with the reduced cultivated land in space. Sustainable development scenario, which takes into account urban development, agricultural development and ecological protection, still has the highest habitat quality index, which shows that the establishment of ecological red line and permanent basic farmland not only helps to protect food security, but also helps to improve regional habitat quality; (3) The influence of land use on habitat quality was successfully evaluated at the regional scale by coupling the FLUS model with the InVEST model. From the perspective of sustainable planning, this coupling framework is helpful to further understand the impacts that land use can have on ecosystem services, and help decision makers to make more sustainable planning plans in regions like YRD around the world.

## Figures and Tables

**Figure 1 ijerph-18-02389-f001:**
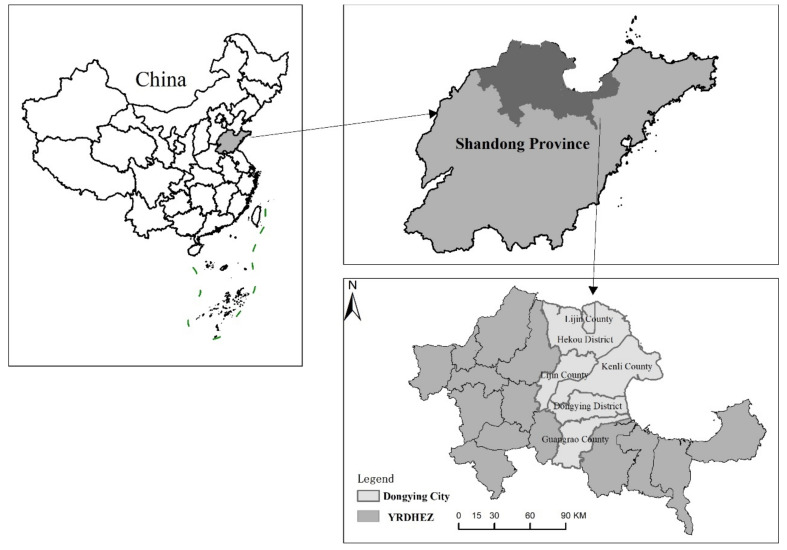
The location of the study ar2.2. Data Resources

**Figure 2 ijerph-18-02389-f002:**
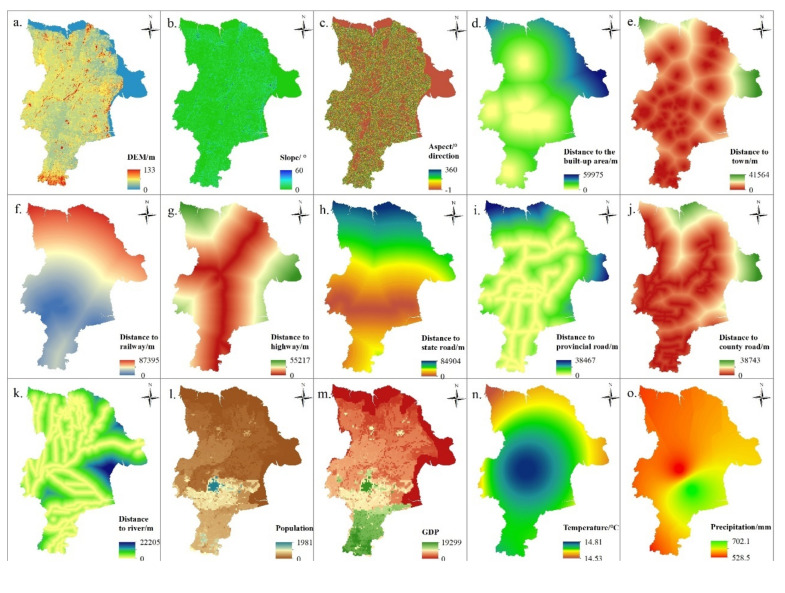
Driving factors of land use in Dongying city ((**a**) elevation; (**b**) slope; (**c**) aspect; (**d**) distance to the built-up area; (**e**) distance to town; (**f**) distance to railway; (**g**) distance to highway; (**h**) distance to state road; (**i**) distance to provincial road; (**j**) distance to county road; (**k**) distance to river; (**l**) population distribution; (**m**) GDP distribution; (**n**) annual average temperature; (**o**) annual average precipitation).

**Figure 3 ijerph-18-02389-f003:**
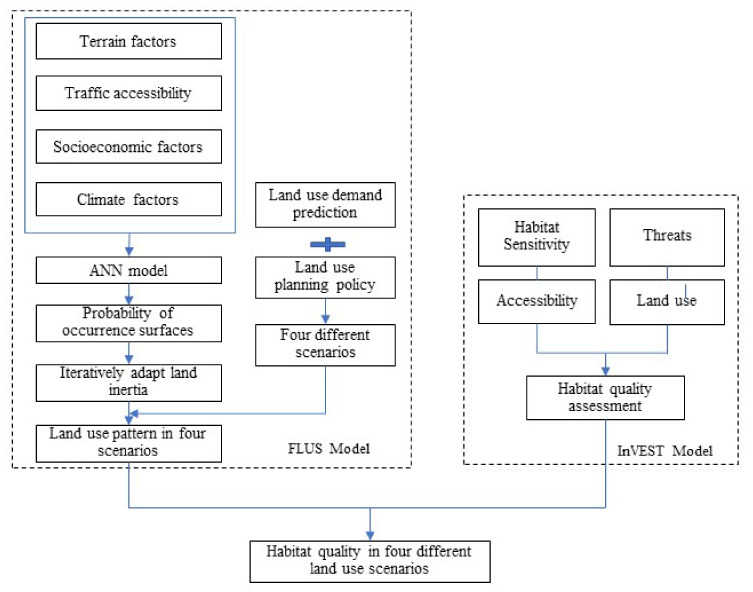
The research framework of FLUS- InVEST models.

**Figure 4 ijerph-18-02389-f004:**
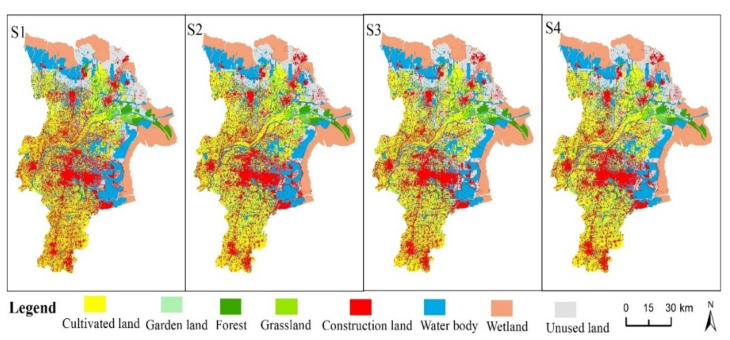
Land use map of 2030 in (**S1**), (**S2**), (**S3**) and (**S4**). Note: S1, S2, S3 and S4 refer to the business as usual scenario, the fast-cultivated land expansion scenario, the ecological security scenario and the sustainable development scenario, respectively.

**Figure 5 ijerph-18-02389-f005:**
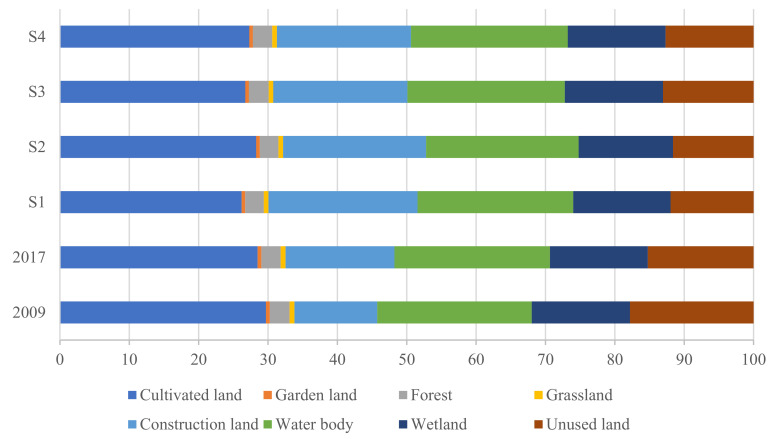
Structure of land use in 2009, 2017 and 2030 under four different scenarios.

**Figure 6 ijerph-18-02389-f006:**
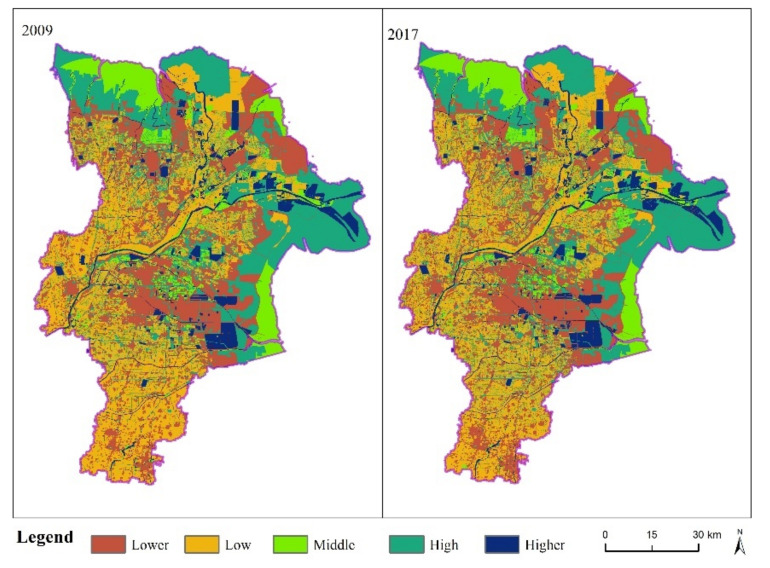
Spatial distribution of habitat quality in Dongying City in 2009 and 2017.

**Figure 7 ijerph-18-02389-f007:**
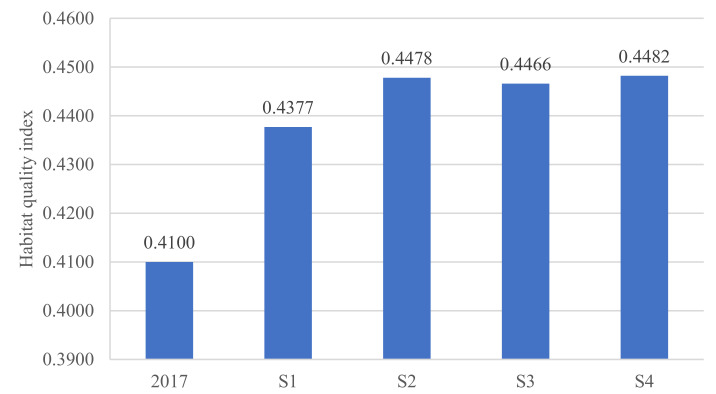
Comparison of habitat quality index in 2017 and 2030 under different scenarios.

**Figure 8 ijerph-18-02389-f008:**
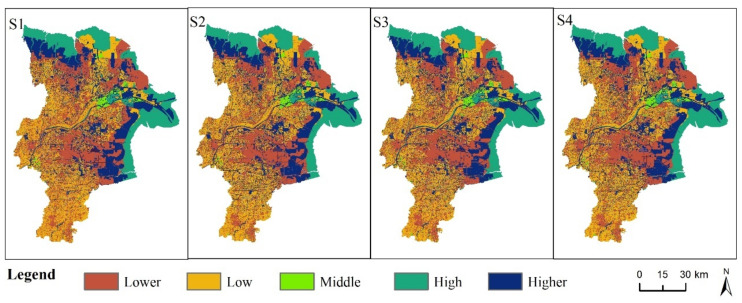
Spatial distribution of habitat quality in Dongying City in (**S1**) (**S2**), (**S3**), and (**S4**). Note: S1, S2, S3 and S4 refer to the business as usual scenario, the fast-cultivated land expansion scenario, the ecological security scenario and the sustainable development scenario, respectively.

**Figure 9 ijerph-18-02389-f009:**
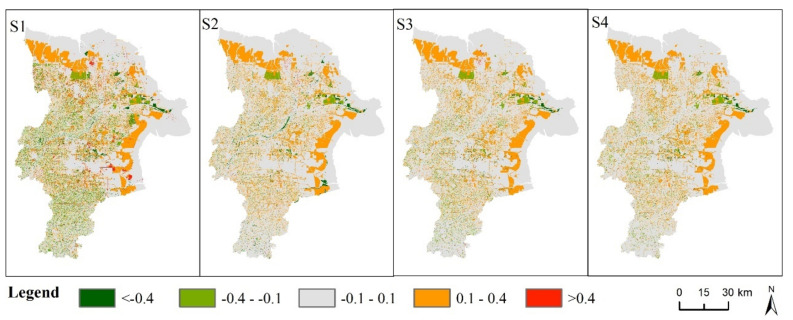
Spatial changes of habitat quality relative to 2017 in Dongying City in (**S1**) (**S2**), (**S3**), and (**S4**). Note: **S1**, **S2**, **S3** and **S4** refer to the business as usual scenario, the fast-cultivated land expansion scenario, the ecological security scenario and the sustainable development scenario, respectively.

**Table 1 ijerph-18-02389-t001:** Data source and relative data descriptions.

Data Type	Data Name	Data Source	Data Description and Procession
Land use	Land use in 2009 Land use in 2017	Natural Resources Bureau	The shape polygon data is converted into raster dataset (30 m× 30 m) using ArcGIS
Planning	Ecological redlines	Natural Resources Bureau	It includes terrestrial ecological redline and marine ecological protection redline. The data is used in the land use scenario simulation as restricted areas.
	Permanent primary farmland	Natural Resources Bureau	It is the highest quality cultivated land in the area. The data is used as planning location factors.
Terrain factors	DEM	Resource and Environment Science Data Center of Chinese Academy of Sciences (http://www.resdc.cn/)	Resolution: 30 m × 30 m. It derived from SRTM data. The slope and aspect data were calculated using ArcGIS.
	Slope		
	Aspect		
Accessibility	Distance to built-up areas	Geographical Information Monitoring Cloud Platform(http://www.dsac.cn/)	The distance data is calculated using ArcGIS Euclidean distance tool.
	Distance to town		
	Distance to railway		
	Distance to highway		
	Distance to state road		
	Distance to provincial road		
	Distance to county road		
	Distance to river		
Socioeconomic factors	Population distribution	Resource and Environment Science Data Center of Chinese Academy of Sciences (http://www.resdc.cn/)	We clipped the spatial distribution data of China’s population and GDP. The resolution was converted to 30 m × 30 m from the 1 km precision.
	GDP distribution		
Climate factors	Annual average temperature	China Meteorological Administration (http://data.cma.cn/)	We used the inverse distance weighted (IDW) interpolation method to spatially interpolate Chinese meteorological data and extracted the meteorological spatial data of Dongying City.
	Annual average precipitation		

**Table 2 ijerph-18-02389-t002:** Parameters and planning policy under four different scenarios.

Scenarios	Name	Parameters and Planning Policy
S1	Business as usual (BAU)	The FLUS model parameters remain unchanged, and the area predicted by the Markov chain is used as the land demand data.
S2	Fast cultivated land expansion scenario (FCLE)	Increase the area of cultivated land and decrease the area of construction land, water body, wetland and unused land. Construction land expansion capacity was adjusted to 0.9.
S3	Ecological security scenario (ES)	Increase the area of cultivated land, water body, wetland and unused land, and decrease the area of construction land. The expansion capacity of cultivated land, construction land, forest, water body and wetland were adjusted to 0.2, 0.9, 0.3, 0.2, 0.2, respectively. Ecological redline is used as a restricted area. The conversion cost matrix was changed, see Table 5 for details.
S4	Sustainable development scenario (SD)	Increase the area of cultivated land, forest, water body and wetland, and decrease the area of construction land. The expansion capacity of cultivated land and construction land were adjusted to 0.4 and 0.9. Ecological redline and permanent prime farmland preservation zone were used as restricted areas. The conversion cost matrix is same as S3 (Table 5).

**Table 3 ijerph-18-02389-t003:** Predicted area of land use types in four planning scenarios in Dongying City in 2030 units:hm^2.^

Scenarios	Cultivated Land	Garden Land	Forest	Grassland	Construction Land	Water Body	Wetland	Unused Land
S1	215,993	4262	22,208	5487	177,100	185,178	115,601	98,814
S2	233,273	4262	22,208	5487	169,957	181,474	112,133	95,849
S3	220,313	4262	23,512	5487	159,390	187,215	116,757	107,707
S4	225,065	4262	23,097	5487	159,390	186,420	116,179	104,743

**Table 4 ijerph-18-02389-t004:** The neighborhood factor parameters.

Scenarios	Cultivated Land	Garden Land	Forest	Grassland	Construction Land	Water Body	Wetland	Unused Land
S1	0.5	0.1	0.2	0.1	1	0.1	0.1	0.1
S2	0.5	0.1	0.2	0.1	0.9	0.1	0.1	0.1
S3	0.2	0.1	0.3	0.1	0.9	0.2	0.2	0.1
S4	0.4	0.1	0.2	0.1	0.9	0.1	0.1	0.1

**Table 5 ijerph-18-02389-t005:** Conversion cost matrix.

Scenarios	Land Use Type	Cultivated Land	Garden Land	Forest	Grassland	Construction Land	Water Body	Wetland	Unused Land
S1 & S2	Cultivated land	1	0	1	0	1	1	0	0
	Garden land	1	1	1	0	1	1	0	0
	Forest	1	0	1	0	1	0	0	0
	Grassland	1	1	1	1	1	1	0	0
	Construction land	1	1	1	0	1	0	0	0
	Water body	1	1	1	0	1	1	1	0
	Wetland	1	1	1	1	1	1	1	0
	Unused land	1	1	1	1	1	1	1	1
S3& S4	Cultivated land	1	0	1	0	1	1	1	0
	Garden land	1	1	1	0	1	1	1	0
	Forest	1	0	1	0	1	1	1	0
	Grassland	1	1	1	1	1	1	1	0
	Construction land	1	1	1	1	1	0	1	0
	Water body	1	1	1	1	1	1	1	0
	Wetland	1	1	1	1	1	1	1	0
	Unused land	1	1	1	1	1	1	1	1

Note: When one type of land use is not allowed to converse to any other, we set the corresponding value of the matrix to 0; set to 1 when the conversion is allowed.

**Table 6 ijerph-18-02389-t006:** Threat factors and weight in the study area.

Threat Type	Max distance	Weight	Decay
Urban land	10	1	exponential
Rural residential land	5	0.7	exponential
Transportation land	2	0.6	linear
Mining land	4	0.8	exponential
Cultivated land	1	0.5	exponential
Saline alkali land	1	0.6	exponential

**Table 7 ijerph-18-02389-t007:** The sensitivity of land use to each threat factor in the study area.

Land Use Type	Habitat Suitability	Urban Land	Rural Residential Land	Transportation Land	Mining Land	Cultivated Land	Saline alkali Land
Cultivated land	0.4	0.5	0.6	0.2	0.5	0.3	0.8
Garden land	0.5	0.4	0.4	0.3	0.4	0.3	0.7
Forest	0.9	0.9	0.8	0.7	0.7	0.5	0.4
Grassland	0.7	0.7	0.4	0.6	0.6	0.4	0.3
Construction land	0	0	0	0	0	0	0
Water body	0.9	0.8	0.6	0.4	0.9	0.6	0.7
Wetland	0.7	0.8	0.6	0.4	0.9	0.7	0.7
Unused land	0.1	0.6	0.6	0.3	0.2	0	0.9

**Table 8 ijerph-18-02389-t008:** Habitat quality grade classification and area proportion in2017 and 2030 (%).

Habitat Quality Grade	Habitat Quality Index	2017	S1	S2	S3	S4
Lower	0–0.1	26.99	31.34	29.14	30.22	29.67
Low	0.1–0.4	30.07	28.41	30.31	28.74	29.32
Middle	0.4–0.6	9.37	1.51	1.57	1.73	1.68
High	0.6–0.7	26.55	14.74	14.29	14.80	14.80
Higher	0.7–1	7.03	23.99	24.70	24.51	24.53

## Data Availability

The data presented in this study are available on request from the corresponding author.

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
