# Peer review of "Multi-Scenario Analysis of Habitat Quality in the Yellow River Delta by Coupling FLUS with InVEST Model"

_ijerph, 2021, doi:10.3390/ijerph18052389_

Round 1

Reviewer 1 Report

It is an interesting and necessary topic given the current situation of ecological and sustainability crisis we are living through. However, I recommend that the authors improve their writing in the following sections:

INTRODUCTION

It is necessary to define what each of the 4 scenarios (characteristics?) refers to.

METHODOLOGY

It is explained how the multi-scenario analysis is carried out (procedure) but it is not clear what this analysis consists of (method?), it is necessary to explain.

RESULTS

There is no correspondence or the relationship between the content and title of Table 8 and what is explained in the text is not well explained.

DISCUSSION AND CONCLUSIONS

I believe that the content of each section should be exchanged between them, in fact the Discussion section begins with "In this paper, we conclude that...".

Reviewer 2 Report

Habitat quality is experiencing unprecedented threats, especially in developing countries. Land use change is one of the most significant factors for the habitat quality change. The authors established a framework by coupling the FLUS model with InVEST model and analyzed the distribution and changes of land use and habitat under various scenarios. The research is very interesting and can provide useful policy suggestions for land use and ecological environment management in the Yellow River Delta. The aim is pertinent and related to the aim of this journal.

I’d suggest minor revisions for this paper to be published in this journal. There are some issues to pay attention to, as follows:

There are three parts that need to be strengthened:

  • In section 2, please explain the reason for the selection of the indicators in the FLUS model.
  • The innovation of this article is not clear. It seems to me that combining the FLUS with InVEST model is the contribution of this paper but the authors should state what is the advantages of doing so.
  • The policy implications are missing. I suggest the authors should discuss it in section 4.

There also some minor problems:

1) Introduction section

I suggest to rewrite long sentences into short sentences, such as “In this paper, we established a framework by coupling the future land use simulation (FLUS) model with the Integrated Valuation of Environmental Services and Tradeoffs (InVEST) model to project the habitat quality change of land use change in Dongying City and compared under scenarios: business as usual (BAU), fast cultivated land expansion scenario (FCLE), ecological security scenario (ES) and sustainable development scenario (SD)

2) Section 2.2

What does the “roadnet work ” mean? Is that “road network”

3) Section 2.2

There are some Chinese labels in Figure 2k.

4) Section 2.3.3

Table 5 and Table 7 are not quoted in the text, table 6 is in the wrong order.

5)In figure 6, the north arrow and legend are too small to see clearly, I suggest to adjust them and check the same problems in other figures.

6) I suggest adjusting the letters on the left side of figure 7.

7) English should be checked through the manuscript.

Reviewer 3 Report

Habitat quality is important for human beings, especially in China, which has paid more attention to environmental and ecological protection since 2015. The paper applied the FlUS and InVEST model to simulate and compare the dynamic change of habitat quality under multi-scenario till 2030 by taking Dongying City as an example.  The research idea is good, and the content is scientific; the results have a certain practical significance. However, there are still several questions left behind that need to be clarified or modified.

First, give more intention to the tense problem in the paper. The tense need to be uniformed;

Second, the YRDHEZ only emerges one time in the context, and it's no need to have an abbreviation about the Yellow River Delta High-efficient Eco-Economic Zone;

Third, in figure 2, the Chinese name in the subgraph of k need to be translated into English, the legend name within the subgraph of d-k need to be more specific, such as change the "Distance" to the "Distance to highway" ;

Fourth, in line 163, the cultivated land increased from 235032 to 233273 hm2, it's strange cause 233273 is less than 235032;

Fifth, in line 166-167, giving the S3 is the ecological security scenario, maybe there is no need to consider the construction land;

Sixth, all formulas in the paper need to be right-aligned;

Seventh, the font size of each table in the paper need to be consistent;

Eighth, in figure 4, the legend, arrow, and scale need to rearrange, and the meaning of the number in the legend should be specific;

Ninth, the land use not only affects habitat quality but also threatens other ecosystem services, such as carbon storage and water conservation. It is also meaningful and feasible to combine the InVEST and FLUS model to study the impact of land use on the other ecosystem services. This is not the limitation. In fact, it's the paper's creation in my mind.

Tenth, in part 4, the discussion could separate into several sections; for example, the first section could discuss the differences of simulation results under four scenarios, the second section could discuss the protest for future habitat protection, and the last section could discuss the limitation and improvement.

In summary, the above-mentioned questions and advice paper is just for reference, and the research context is suitable. 
